# Nitroglycerin (NTG) Infusion for Intraprocedural Vasospasm in Transarterial Microembolization (TAME): A Case Series

**DOI:** 10.3390/life14111413

**Published:** 2024-11-01

**Authors:** Kuan-Wei Li, Keng-Wei Liang, Wen-Ying Liao, Chien-Kuo Wang, Yi-Sheng Liu, Tai-Hua Yang, Chun-Hsin Wu, Bow Wang

**Affiliations:** 1Department of Medical Imaging, National Cheng Kung University, Tainan 704, Taiwan; sinhcoshverysinh@gmail.com (K.-W.L.); n625396@mail.hosp.ncku.edu.tw (C.-K.W.); n041075@mail.hosp.ncku.edu.tw (Y.-S.L.); 2Department of Medical Imaging, Chung Shan Medical University Hospital, Taichung 402, Taiwan; james095797@gmail.com; 3School of Medicine, Chung Shan Medical University, Taichung 402, Taiwan; 4Department of Biotechnology and Bioindustry Sciences, National Cheng Kung University, Tainan 704, Taiwan; wyliao777@gmail.com; 5Department of Biomedical Engineering, National Cheng Kung University, Tainan 704, Taiwan; yangtaihua@mail.ncku.edu.tw; 6Department of Orthopedic Surgery, National Cheng Kung University Hospital, Tainan 704, Taiwan; 7Medical Device Innovation Center, National Cheng Kung University, Tainan 704, Taiwan; 8Department of Internal Medicine, Division of Allergy, Immunology, and Rheumatology, National Cheng Kung University Hospital, College of Medicine, National Cheng Kung University, Tainan 704, Taiwan; hsinjo@gmail.com; 9Interventional Medicine Center, National Cheng Kung University Hospital, College of Medicine, National Cheng Kung University, Tainan 704, Taiwan

**Keywords:** intraprocedural vasospasm, vasospasm, NTG, TAME, muscular artery

## Abstract

Transarterial microembolization (TAME) is an innovative approach to treating chronic musculoskeletal pain. During the procedure, intraprocedural vasospasm, presenting juxta-catheter stenosis, and poor distal artery perfusion and flow through the anastomosis commonly pose challenges. Difficulty of the catheter reaching the target vessel and restricted drug delivery are possible consequences. To address these issues, transcatheter nitroglycerin (NTG) infusion at the extremity’s small-sized artery has been introduced. We investigated patients who underwent the TAME procedure wherein NTG was employed and sourced from two institutional settings. Transcatheter NTG infusion was administered in seven instances of chronic musculoskeletal pain manifesting with intraprocedural vasospasm during TAME procedures. Patient profiles, preprocedural imaging, fluoroscopic findings, adverse events, and Numerical Rating Scale (NRS) scores were evaluated. As a result, all seven cases experiencing intraprocedural vasospasm exhibited rapid responses to transcatheter NTG infusion. Angiography conducted pre- and post-infusion demonstrated increased vessel size, resolved proximal catheter stenosis, and improved distal perfusion. One case presented an adverse effect of self-limited transient hypotension. The NRS scores decreased following the procedure. Transcatheter NTG infusion at the extremity’s small-sized muscular artery can be an effective technique for resolving intraprocedural vasospasm in TAME procedures, irrespective of the target diseases and approach vessels.

## 1. Introduction

Transarterial microembolization (TAME) represents an emerging treatment modality for chronic musculoskeletal pain [1,2,3]. Angiography is conducted at the main feeding vessel or the parent vessel of the targeting tissue to identify inflammatory neo-vessels. Selective angiography is performed at each targeting vessel, followed by embolization to reduce blood flow to the inflammatory neo-vessels.

Intraprocedural vasospasm is a transient stenosis induced by mechanical stimulation from a guidewire or catheter [4,5], and is a common challenge during TAME procedures [6,7]. Small muscular arteries are particularly vulnerable to vasospasm due to their significant amount of smooth muscle content [8]. Consequences of intraprocedural vasospasm include difficulty in catheter navigation to the target vessel [9] and restricted drug delivery. Vasospasm is identified on digital subtraction angiography (DSA) by comparing the vascular contour with the angiography of proximal vessels and normal anatomy. Manifestations of intraprocedural vasospasm include proximal catheter stenosis, marked by an abrupt focal or short segment decrease in vessel size, as in Figure 1, and impaired distal perfusion, presented as limited opacified soft tissue staining on selective angiography, as in Figure 2. Additionally, vasospasm in areas of anastomosis may cause the enhancement of anastomotic vessels, as in Figure 3, diverting embolic agents away from their intended treatment targets [10]. Vasospasms encountered during the TAME procedure have been reported to resolve after approximately 30 min of waiting [6,7]. To address the challenges of vasospasms, transcatheter nitroglycerin (NTG) infusion is introduced as a potential for a time-efficient, treatment-augmentation solution.

NTG, a short-acting vasodilator, is widely used in various clinical settings to treat vasospasms in digital arteries [11], the radial artery [12,13], and radial artery graft of the coronary artery [14]. Additionally, NTG is employed to augment drug delivery [15]. Notably, it is used in transarterial embolization or transarterial chemoembolization for the treatment of hepatic tumors to enhance the delivery of embolic agents [16,17]. This study aims to introduce the NTG infusion technique for alleviating intraprocedural vasospasms, assessing the efficacy and safety of perfusion modification, and evaluating the post-treatment response.

## 2. Materials and Methods

### 2.1. Patient Selection

Our study adopts a retrospective approach, focusing on patients with chronic musculoskeletal pain who underwent the TAME procedure, utilizing intraprocedural NTG sourced from two institutional settings between April 2020 and May 2024. Transcatheter NTG infusion was administered to manage intraprocedural vasospasm during TAME procedures. Specifically, this study included seven cases of chronic musculoskeletal pain, lateral epicondylitis, post-traumatic foot pain, triangular fibrocartilage complex (TFCC) tear, finger osteoarthritis (OA), patellar tendinopathy, metatarsal refractory pain, and secondary stiff shoulder, as in Table 1. All seven patients were checked for normal renal functions (eGFR > 90) for the safety of transarterial angiography. Clinical records and pre- and intra-TAME images were reviewed, including assessments of patient characteristics, preprocedural imaging diagnosis, DSA imaging assessments, adverse events, and evaluations of treatment outcomes.

Patient characteristics include age, sex, symptom duration, pain scores based on numerical rating scale (NRS) scores, and previous treatments (rehabilitation, prolotherapy, local steroid injection, and surgery), as listed in Table 2. Preprocedural images were assessed to establish the symptom–image correlation. Plain radiographs and MRIs were reviewed for accurate diagnosis.

### 2.2. Preparation of NTG

The preparation of NTG involved mixing 500 mcg of NTG (Millisrol 5 mg/10 mL/amp, Nippon Kayaku Co., Ltd., Tokyo, Japan) with normal saline to a total volume of 10 mL.

### 2.3. DSA Imaging Assessment and Analysis

The fluoroscopy images of all enrolled patients were evaluated, including digital subtraction angiography (DSA) of the parent vessels, targeted vessels, and treatment-related angiographies. Quantitative and qualitative assessments of the spastic vessels were performed on both pre- and post-NTG images. The DSA of the parent vessels was reviewed to compare the spastic vessels on the DSA of the main feeding vessels to the pre- and post-NTG images. A vasospasm was identified by comparing the vascular contour with the angiography of the parent vessels and normal anatomy. It was described quantitatively by decreased vascular size and qualitatively by vasospasm manifestations, including proximal catheter stenosis, impaired distal perfusion, and flow through anastomotic vessels. Post-TAE images were also assessed to evaluate the success of the procedure, as shown in Figure 1, Figure 2, Figure 3, Figure 4, Figure 5 and Figure 6.

### 2.4. Quantitative Measurements

Vessel size was measured on both the pre- and post-NTG DSA images. The image that showed the most distal opacification of the vasculature was selected for measurement in each DSA. The vascular size on the pre-NTG images was measured at the narrowest point of stenosis if proximal catheter stenosis was noted; otherwise, a segment of the spastic vessel before branching was measured. The vascular size on the post-NTG images was measured at the same level as on the corresponding pre-NTG images.

### 2.5. Statistical Analysis

A small-sized patient group was included in this study, with variable vascular anatomy among these cases. Thus, the Wilcoxon signed rank exact test was applied to the comparison of vascular sizes between pre-NTG images and post-NTG images. The significance level was set as *p* < 0.05.

### 2.6. Qualitative Assessments

In each case, the vasospasm manifestations were assessed. Proximal catheter stenosis was identified as a marked abrupt decrease in vessel size. Impaired distal perfusion was identified as limited opacified soft tissue staining on selective angiography. Flow through anastomotic vessels was identified with the aid of the vascular anatomy of the target vessel.

### 2.7. Adverse Events

Following a previous report [12] and pharmacology instructions, intraprocedural blood pressure, oxygen saturation, and patient symptoms were recorded as a clinical routine. The medical recording of procedures was assessed for intraprocedural hypotension and other adverse events.

### 2.8. Evaluations of Treatment

The success of vasospasm relief was considered achieved if the distal abnormal soft tissue staining was presented after the NTG infusion. Complete embolization was considered if the abnormal soft tissue staining was reduced after the embolization. Post-TAME pain was recorded using NRS scores.

### 2.9. TAME Procedure

All procedures were performed by two interventional radiologists (B.W., and K.-W.L.) with over 5 years of expertise.

TAE procedures were performed under local anesthesia, with percutaneous arterial access obtained using a 4-Fr. introducer sheath (Merit Medical) via the entry artery. An angiocatheter (J curve small tip, 4 Fr. 65 cm, Terumo, Terumo Vietnam Co., Ho Chi Minh City, Vietnam; Judkins Right 4 Fr. 100 cm, Terumo, Terumo Vietnam CO, Vietnam, or RIM 5 Fr. 100 cm, Cook Medical, Bloomington, IN, USA) was used to approach the parent vessel as well as perform angiography to identify the anatomy of the orifices of targeting vessels. A 1.98 Fr (Asahi Masters Parkway Soft, Asahi Intecc Co., Ltd., Chonburi, Thailand.). or 1.7 Fr. microcatheter (Lambda microcatheter, Terumo, Terumo Corporation, Japan or Veloute microcatheter, Asahi INTECC, Thailand, Co., Ltd) was used to selectively approach the target vessels. A selective DSA was performed at each targeting vessel.

### 2.10. NTG Infusion

When an intraprocedural vasospasm was identified on the DSA, NTG was then ordered. The preparation of NTG involved mixing 500 mcg of NTG (Millisrol 5 mg/10 mL/amp, Nippon Kayaku Co., Ltd.) with normal saline to a total volume of 10 mL. Then, 1–2 ml of diluted NTG (containing 50–100 mcg) was administered via the microcatheter. After a brief 30–60 s of waiting time, a selective DSA was performed. With the presentation of abnormal inflammatory neo-vessels, embolization was performed using a suspension mixture of 500 mg of imipenem/cilastatin (IPM/CS) in 10 mL of iodinated contrast as the embolic agent based on previous reports [3,18,19].

## 3. Detailed Case Description

### 3.1. Lateral Epicondylitis

A 61-year-old woman presented to the clinic with left chronic elbow pain and range of motion (ROM) limitation and reported a history of previous treatment of surgery and needle acupuncture.

An MRI of the left elbow showed tendinopathy of the common extensor tendon at the insertion of the lateral humeral epicondyle. TAME was arranged.

During the procedure, as in Figure 1, an angiography at the left radial recurrent artery showed a vasospasm with a proximal catheter stenosis point. An NTG infusion was given, and the vascular stenosis was completely resolved, measuring from 0.99 mm to 2.45 mm. The NRS improved from 7 to 4 at 1 month, 3 at 3 months, and 2.5 at 6 months follow-up. The QuickDash scores were improved from 52.3 as baseline, 29.5 at 1 month, 15.9 at 3 months, and 11.4 at 6 months follow-up.

### 3.2. Post Traumatic Foot Pain

A 26-year-old man presented to the clinic with bilateral forefoot plantar side pain at the 2nd-4th toes for three years after a car accident and reported a history of previous treatment of bilateral forefeet platelet-rich plasma (PRP) injection, medication, and rehabilitation without obvious improvement.

Plain radiography of the left foot showed no fracture or dislocation, and an MRI showed no obvious tendon or ligament tears. TAME was arranged.

During the procedure, as in Figure 2, angiography at the left arcuate artery showed vasospasm with impaired perfusion of left dorsal metatarsal arteries and proximal catheter stenosis. An NTG infusion was given, which improved perfusion, and the vascular stenosis was completely resolved, measuring from 1.75 mm to 2.46 mm. However, hypotension was noted after the NTG infusion. The procedure was then temporarily ceased, and a normal saline drip was given. The blood pressure of the patient recovered in a few minutes. The TAME procedure was then continued. The NRS improved from 10 to 6 at 1 month, 5 at 3 months, and 5 at 6 months follow-up.

### 3.3. Triangular Fibrocartilage Complex (TFCC) Tear

A 43-year-old woman presented to the clinic with pain at the ulnar side of the right wrist for two years and reported a history of previous treatment of long-term physical therapy and shock wave therapy without significant improvement. She undertook local injection with prolotherapy three times and a steroid injection, which resulted in partial improvement.

An MRI of the right wrist revealed a torn TFCC of the right wrist. TAME was arranged.

During the procedure, an angiography at the right ulnar artery showed vasospasm with impaired perfusion to the TFCC and proximal catheter stenosis. An NTG infusion was given, which improved perfusion, and the vascular stenosis was completely resolved. The NRS was improved from 7 to 5 at 1 month, 4 at 3 months, and 3 at 6 months follow-up. The Patient-Rated Wrist Evaluation (PRWE) scores were improved from 48 as a baseline, 27.5 at 1 month, 23 at 3 months, and 25 at 6 months follow up.3.4 Finger osteoarthritis (OA)

A 55-year-old woman presented to the clinic with left distal interphalangeal (DIP) joint pain and swelling for two years.

Plain radiography showed joint space narrowing with spur formation in multiple DIP joints. TAME was arranged.

During the procedure, as in Figure 3, an angiography at the left radial artery showed vasospasm with flow through the superficial palmar arch via anastomosis, proximal catheter stenosis, and impaired perfusion of the deep palmar arch with the absence of blood flow at the DIP joint level. An NTG infusion was given, which improved perfusion, and the vascular stenosis was completely resolved, measuring from 1.12 mm to 2.71 mm. The NRS improved from 7 to 3 at 1 month, 2 at 3 months, and 1 at 6 months follow-up. The Functional Index of Hand Osteoarthritis (FIHOA) scores were improved from 21 as a baseline, 11 at 1 month, 6 at 3 months, and 6 at 6 months follow-up.

### 3.4. Patellar Tendinopathy

A 36-year-old woman presented to the clinic with left knee pain after a traffic accident for one year and reported a history of previous treatment of glucose, PRP injection for anterior cruciate ligament (ACL), patellar tendon, and medial meniscus, and shock wave therapy three times for patella tendon, resulted in no significant improvement.

An MRI of the right knee showed left patellar tendinopathy at both patellar and tibial insertion sites. TAME was arranged.

During the procedure, as in Figure 4, angiography at the left lateral inferior genicular artery showed flow through the left medial superior genicular artery via anastomosis, impaired perfusion, and proximal catheter stenosis. An NTG infusion was given, which improved perfusion of left lateral inferior genicular artery territories, and the vascular stenosis was resolved, measuring from 0.85 mm to 1.03 mm. The anastomotic flow to the left medial superior genicular artery was not seen. The NRS was improved from 6.5 to 4 at 1 month, 2 at 3 months, and 1.5 at 6 months follow-up.

### 3.5. nd-4th Metatarsalgia

A 20-year-old man presented to the clinic with bilateral foot 2nd-4th metatarsalgia for two years and reported a history of previous treatment of rehabilitation and PRP injection twice, which resulted in partial improvement.

Plain radiography showed unremarkable bony structures in both feet. TAME was arranged.

During the procedure, as in Figure 5, angiography at the right anterior tibial artery showed vasospasm with impaired perfusion to the right dorsal metatarsal arteries. An NTG infusion was given, which improved perfusion to the right dorsal metatarsal arteries. The NRS was improved from 10 to 5 at 1 month, 5 at 3 months, and 4 at 6 months follow-up.

### 3.6. Secondary Stiff Shoulder

A 62-year-old woman presented to the clinic with right shoulder pain during arm elevation and put down for about 2 years post operation of repairment of rotator cuff tear. She reported a history of previous treatment of local steroid injection twice and prolotherapy, which resulted in partial improvement.

Ultrasonography showed increased echogenicity and vascularity over the coracoid area. An MRI of the right shoulder showed postoperative changes with a metallic artifact, partial-thickness tear of the supraspinatus tendon, biceps tendon long-head tendinopathy, and osteoarthritis of the right acromioclavicular joint. TAME was arranged.

During the procedure, as in Figure 6, angiography at the right thoracoacromial artery showed vasospasm with proximal catheter stenosis and impaired distal perfusion distal to the stenosis point. An NTG infusion was given, which improved perfusion, and the vascular stenosis was completely resolved, measuring from 1.12 mm to 1.96 mm. The NRS did not change from 7 to 7 at 1 month, 7 at 3 months, and 7 at 6 months follow-up. The QuickDash scores were improved from 77.3 as baseline, 68.2 at 1 month, 79.5 at 3 months, and 77.3 at 6 months follow-up.

## 4. Results

### 4.1. Quantitative Changes in Vascular Size

Vasospasms were observed in small-sized muscular arteries, with diameters ranging from 1.03 to 3.77 mm on post-NTG images, as shown in Table 1. In the angiographic studies where vasospasm was detected, five out of seven cases (71.4%) exhibited proximal catheter stenosis. The diameters in these seven cases ranged from 0.85 to 3.02 mm before NTG infusion and from 1.03 to 3.77 mm post-NTG infusion (*p* < 0.05), as detailed in Table 1.

### 4.2. Qualitative Vasospasm Manifestations

For the qualitative assessment of vasospasm manifestations, as shown in Table 3, six out of seven cases (85.7%) exhibited impaired distal perfusion, all of which resolved following NTG infusion. Additionally, three of the seven cases (42.9%) demonstrated flow through anastomotic vessels, but in two of these cases, the anastomotic vessels were no longer opacified after NTG infusion.

### 4.3. Patients Outcomes

The patients who received an NTG infusion experienced promising outcomes. All patients showed relief from vasospasm, as evidenced by the abnormal vascular staining corresponding to inflammatory neo-vessels that appeared after NTG infusion. Additionally, embolization was successfully completed in all NTG-treated vessels. The average NRS score decreased from 7.8 before treatment to 3.4 at 6 months follow-up after treatment (Table 4). One patient (14%) experienced hypotension during the procedure following NTG infusion, but the blood pressure recovered within a few minutes after intravenous fluid administration.

## 5. Discussion

### 5.1. Efficacy and Safety

Intraprocedural vasospasm has been identified as a common challenge during TAME procedures [7]. During selective angiography at the level of the muscular arteries, the movement of the guidewire and the relative size of the catheter compared to the vessel can cause significant stimulation to the target vessel. Vasospasm, which presents as proximal catheter stenosis, impaired distal perfusion, or flow through anastomotic vessels, hinders the operator’s ability to properly visualize the target vascular anatomy and pathology. Additionally, disturbed flow due to vasospasm limits the effective delivery and distribution of the embolic agent.

Nitroglycerin, by acting on smooth muscle cells at the intracellular level, causes the synthesis of guanosine 3′,5′-monophosphate in smooth muscle cells, stimulating a guanosine 3′,5′-monophosphate–dependent protein kinase with resultant dephosphorylation of the light chain of myosin and relaxation of the contractile state of smooth muscle, causes rapid and short-term relaxation of vascular tone. Intra-arterial infusion of NTG, typically ranging from 50 to 500 mcg, has been reported in various intervention studies [11,12,16]. In this study, intraprocedural DSA performed before and after NTG infusion consistently showed improvement in vasospasm. The procedure was completed with a brief waiting time of 30 s to 1 min, and the delivery of imipenem/cilastatin was optimal in all cases. Assessments using NRS scores further confirmed the procedure’s effectiveness for pain relief.

Subsequent post-procedural analysis conducted using the Picture Archiving and Communication System (PACS) revealed both quantitative and qualitative improvements in vasospasm, confirming the efficacy of the intra-arterial NTG infusion.

In contrast to previous studies where intra-arterial NTG has been associated with adverse events, such as hypotension and headache [12], our study observed one NTG-related adverse event as transient hypotension. The procedure was temporarily halted, and intravenous fluids were administered. The patient’s blood pressure recovered within a few minutes.

Consequently, transcatheter NTG infusion is regarded as a time-saving intervention while maintaining or enhancing treatment efficacy and as a rather safe procedure with acceptable adverse effects and rather accessible management.

### 5.2. Prevention of the Vasospasm

As seen in (Table 4), 71% of vasospasms present proximal stenosis of the guiding catheter. The stimulation from the microcatheter and guidewire is thus considered responsible for this type of presentation. Administering embolic material may reduce the mechanical stimulation and prevent the vasospasm.

However, there are some remarkable benefits in selective microembolization, such as better evaluation of the inflammatory neo-vessels before and after the treatment, avoiding flow diversion toward the nontargeting vessels during embolization and achieving better presentation of the provoking pain during treatment, and better dosage adjustment. The non-selective microembolization is thus not generally applied in all TAME cases.

Despite the promising findings, this study has several limitations that should be considered when interpreting the results. First, the retrospective nature of the study inherently carries potential biases in patient selection and data collection, as we relied on existing medical records and imaging studies. Second, the sample size is small, with only seven cases included, which limits the generalizability of our findings and the statistical power of our analyses. Third, there were variations in the treated vessels, including differences in vessel size and anatomical location, which may affect the consistency of the results and the reproducibility of the procedure. Further prospective studies with larger sample sizes and standardized treatment protocols are needed to validate our findings and optimize the management of intraprocedural vasospasm during TAME procedures.

## 6. Conclusions

In conclusion, the NTG infusion method demonstrated in this study shows promise as a time-efficient treatment-augmentation approach for managing intraprocedural vasospasm during the TAME procedure.

The NTG infusion successfully alleviated vasospasm, as evidenced by improved vascular visualization and the successful delivery of embolic agents. Additionally, the short waiting time and lack of major adverse effects highlight the potential of NTG infusion to safely enhance procedural efficiency.

## Figures and Tables

**Figure 1 life-14-01413-f001:**
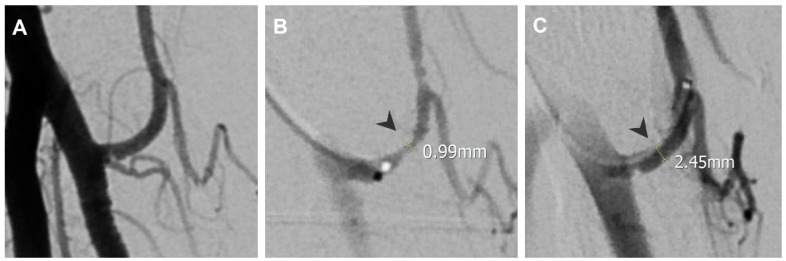
A 61-year-old female with left lateral epicondylitis. (**A**) Angiography at the left brachial artery showing patent left radial recurrent artery without stenosis. (**B**) Angiography at the left radial recurrent artery showing a vasospasm with a proximal catheter stenosis point, measuring 0.99 mm at the narrowest point (black arrowhead). (**C**) Angiography after NTG infusion showing resolved stenosis (black arrowhead, measured 2.45 mm).

**Figure 2 life-14-01413-f002:**
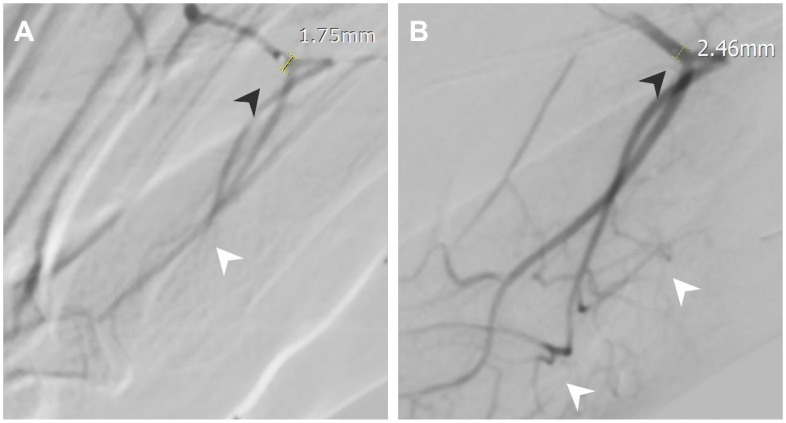
A 26-year-old male with bilateral forefoot plantar side pain at the 2nd-4th toes. (**A**) Angiography at the left arcuate artery showing a vasospasm with impaired perfusion of left dorsal metatarsal arteries (white arrowheads) and proximal catheter stenosis (black arrowheads, 1.75 mm). (**B**) Angiography after NTG infusion showing improved perfusion of left dorsal metatarsal arteries (white arrowheads) and improved proximal catheter stenosis (black arrowheads, 2.46 mm).

**Figure 3 life-14-01413-f003:**
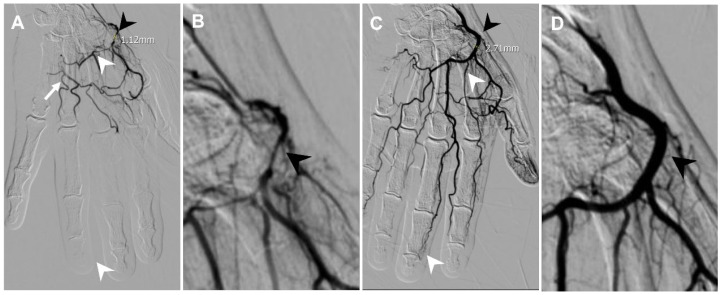
A 55-year-old female presented with pain and swelling at the DIP joints of their left hand for 2 years. (**A**) Angiography at the left radial artery showing a vasospasm with flow through the superficial palmar arch via anastomosis (white arrow), proximal catheter stenosis (black arrowhead, measured 1.12 mm), and impaired perfusion of the deep palmar arch with absence of blood flow at the DIP joint level (white arrowheads). (**B**) Magnification of proximal catheter stenosis (black arrowhead). (**C**) Angiography after NTG infusion showing improved perfusion toward digital arteries at the DIP joint level via the deep palmar arch (white arrowheads). The vascular stenosis was completely resolved (black arrowheads, 2.71 mm). (**D**) Magnification of resolved stenosis (black arrowhead).

**Figure 4 life-14-01413-f004:**
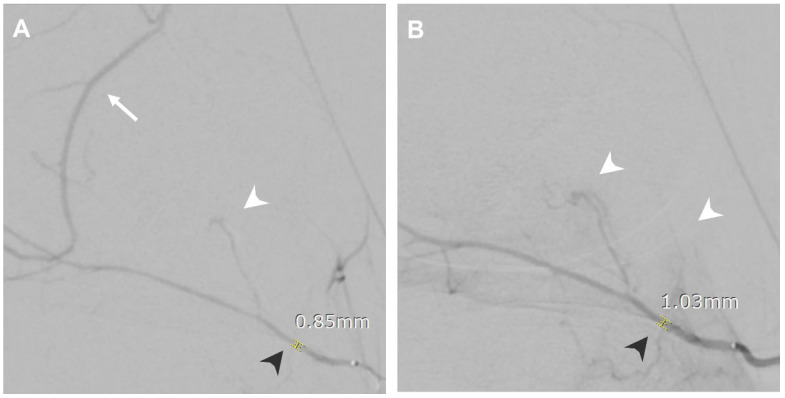
A 36-year-old female presented with left knee pain after a traffic accident for 1 year. (**A**) Angiography at the left lateral inferior genicular artery showing flow through the left medial superior genicular artery via anastomosis (white arrow), impaired perfusion (white arrowhead), and proximal catheter stenosis (black arrowhead, 0.85 mm). (**B**) Angiography after NTG infusion showing improved perfusion of left lateral inferior genicular artery territories (white arrowheads), and the vascular stenosis was resolved (black arrowhead, 1.03 mm).

**Figure 5 life-14-01413-f005:**
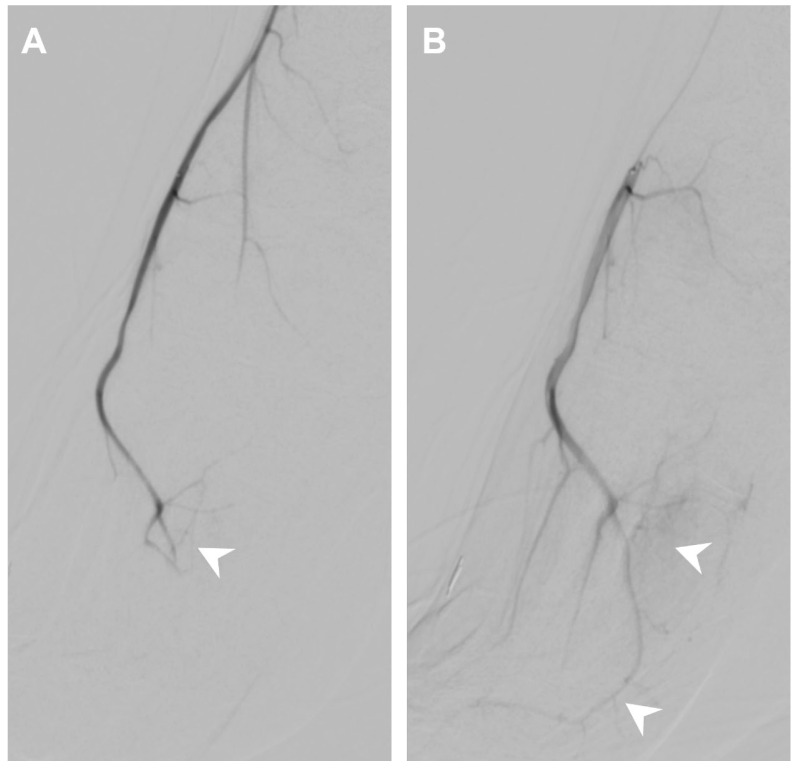
A 20-year-old male presented with bilateral foot 2nd-4th metatarsalgia for 2 years. (**A**) Angiography at the right anterior tibial artery showing vasospasm with impaired perfusion to the right dorsal metatarsal arteries (white arrowhead). (**B**) Angiography after NTG infusion showing improved perfusion to the right dorsal metatarsal arteries (white arrowheads).

**Figure 6 life-14-01413-f006:**
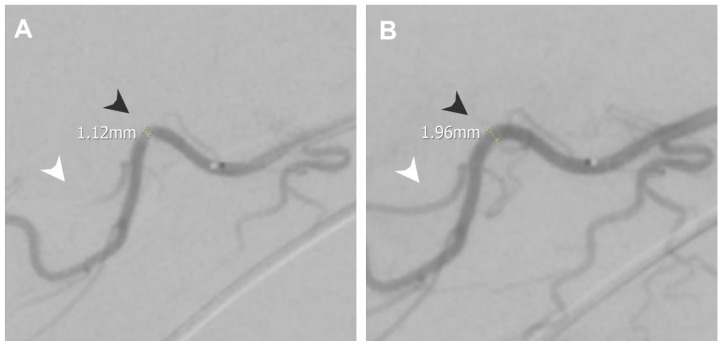
A 62-year-old female presented with right shoulder pain during elevation and put down for about 2 years post-operation. (**A**) Angiography at the right thoracoacromial artery showing vasospasm with proximal catheter stenosis (black arrowhead, measured 1.12 mm) and impaired distal perfusion distal to the stenosis point (white arrowhead). (**B**) Angiography after NTG infusion showing resolved stenosis (black arrowhead, measured 1.96 mm) and improved distal perfusion (white arrowhead).

**Table 1 life-14-01413-t001:** Quantitative changes in vascular size.

Diagnosis	Spasm Vessel	Before NTG (mm)	Post NTG (mm)
Lateral epicondylitis	Radial recurrent artery	1.00	2.5
Post-traumatic foot pain	Arcuate artery	1.75	2.46
TFCC tear	Ulnar artery	2.44	3.14
Finger OA	Radial artery	1.12	2.71
Patellar tendinopathy	Lateral inferior genicular artery	0.85	1.03
2nd–4th metatarsalgia	Anterior tibial artery	3.02	3.77
Secondary stiff shoulder	Thoracoacromial artery	1.12	1.96
			*p* < 0.05

NTG, nitroglycerin; TFCC, triangular fibrocartilage complex; OA, osteoarthritis. *p* value < 0.05 in Wilcoxon signed rank exact test.

**Table 2 life-14-01413-t002:** Patient Characteristics.

Patient Characteristics	Number/Mean
Number	7
Age, years	43.2 ± 16.8
Sex (female vs. male)	5:2 (71%:29%)
Symptom duration, months	37.3 ± 37.1
Laterality (right vs. left)	4: 3 (57%:43%)
NRS score of pain. 0–10	7.8 ± 1.5
Prior treatment	
Rehabilitation	3/7 (43%)
Prolotherapy	5/7 (71%)
Local steroid injection	2/7 (29%)
Surgery	1/7 (14%)

Data are presented as mean ± SD or counts and percentages. NRS, numerical rating scale.

**Table 3 life-14-01413-t003:** Qualitative vasospasm manifestations.

	Before NTG	Post NTG
Proximal catheter stenosis	5/7 (71%)	0/7 (0)
Impaired distal perfusion	6/7 (86%)	0/7 (0)
Flow through anastomotic vessels	3/7 (43%)	1/7 (14%)

NTG, nitroglycerin.

**Table 4 life-14-01413-t004:** Patients Outcomes.

Patient Outcomes	Number/Mean
Vasospasm relief	7/7 (100%)
Complete of embolization	7/7 (100%)
NRS score, 0–10	3.4 ± 2.1
Adverse effects of NTG	
Hypotension	1/7 (14%)
Headache	0/7 (0)
Hematoma	0/7 (0)

Data are presented as mean ± SD or counts and percentages. NRS, numerical rating scale.

## Data Availability

The original contributions presented in the study are included in the article, further inquiries can be directed to the corresponding author.

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
