# Peer review of "Nitroglycerin (NTG) Infusion for Intraprocedural Vasospasm in Transarterial Microembolization (TAME): A Case Series"

_life, 2024, doi:10.3390/life14111413_

Round 1

Reviewer 1 Report

Comments and Suggestions for Authors

Thank you very much for the opportunity to review this valuable manuscript.

The purpose of the manuscript is to report transcatheter nitroglycerin infusion during transarterial microembolization to treat intraprocedural vasospasm. Both pre- and post- NTG DSA images were retrospectively evaluated to measure vessel size change. The methodology was rigorous and the results were well presented and the conclusions seem consistent.

However, during the literature search, several works described the same procedure in different contexts and my concerns are focused on the interest of the readers. Tables and figures explain the results well. References seem appropriate. English language is excellent and few errors were detected:

Line 51 and 53: "as" in figure 1-2 sounds better.

Line 91: between saline and to, space has to be deleted.

In 2.9 paragraph: details about artery access and complication would be beneficial.

Comments on the Quality of English Language

English language is excellent and few errors were detected:

Line 51 and 53: "as" in figure 1-2 sounds better.

Line 91: between saline and to, space has to be deleted.

In 2.9 paragraph: details about artery access and complication would be beneficial.

Author Response

Dear reviewer, thanks for your insightful comments, please see the attached PDF profile and our responses below:

Comments1: Line 51 and 53: "as" in figure 1-2 sounds better.
Response 1: Thank you for pointing this out. We have revised the text on [Page 2, paragraph 1, Lines 51 and 52].

Comments2: Line 91: between saline and to, space has to be deleted.
Response 2: Thank you for pointing this out. The typographical error is corrected on [Page 2, paragraph 2.2, Line 93].

Comments3: In 2.9 paragraph: details about artery access and complication would be beneficial.
Response 3: We appreciate your opinion and have revised the method as stated on [Page 4, Paragraph 2.9, Lines 140-145].

Reviewer 2 Report

Comments and Suggestions for Authors

 This report, which presents a case series of seven patients, is highly valuable as it effectively demonstrates the potential of TAME, an innovative treatment for pain, and provides a rationale for using NTG during the procedure. As the author notes, vasospasm is a common issue encountered during TAME procedures, especially when dealing with small vessels. It is particularly noteworthy that the report suggests NTG can resolve vasospasm quickly without compromising the efficacy of TAME itself. However, I have a few questions and some major revisions that I believe should be addressed, as outlined below.

Major Revisions and Comments

            1.         As you are aware, the effects of TAME are gradual, with pain relief typically occurring over the course of several weeks or months. To avoid any potential misunderstanding, please clarify the time frame of improvement in the NRS, indicating how long the observed benefits lasted.

            2.         The cases included in this report are varied. Among the cases, did you have cases that can provide additional scoring metrics beyond NRS (for pain) to further demonstrate improvements in ADL (activities of daily living) following TAME?

            3.         The target vessels in TAME procedures are often very small, and the use of 4Fr guiding catheters, along with the complexity of the procedure, can sometimes result in vasospasm. Preventive strategies, such as using a 3Fr guiding catheter with a gentler approach and administering embolic material more proximally before reaching the target, have been suggested (Ref 3: J Vasc Interv Radiol, 2022. 33(12): p. 1468-1475.e8). Since you mention that approximately 70% of vasospasms occurred in the proximal part of the guiding catheter, it seems some cases could have been prevented by these techniques, potentially avoiding the need for NTG. This aspect should be discussed further in the manuscript.

            4.         In the seventh case involving a stiff shoulder, did you perform any embolization beyond the right thoracoacromial artery? For conditions like frozen shoulder, embolization of six vessels around the shoulder is recommended (Ref 3: J Vasc Interv Radiol, 2022. 33(12): p. 1468-1475.e8). If the necessary vessels were not targeted, please provide additional details, as this might explain why TAME was ineffective in this case.

Author Response

Thank you for your insightful comments. Please see the attached PDF profile and our responses below:

Comments1: As you are aware, the effects of TAME are gradual, with pain relief typically occurring over the course of several weeks or months. To avoid any potential misunderstanding, please clarify the time frame of improvement in the NRS, indicating how long the observed benefits lasted.
Response 1: Thank you for pointing this out. We totally agree that the effects of TAME are better seen in a series of short- and long-term follow up NRS. The 1 month, 3 months and 6 months follow up NRS were revised in the case description in [Page 4-8, Line 165-166, 187-188, 205, 218, 242-243, 260, 278-279]. In this report, we demonstrated the post treatment NRS at 6 months follow up after the TAME procedure in the patient outcome in [Page 10, Paragraph 4.3, Line 313, Table 4].

Comments 2: The cases included in this report are varied. Among the cases, did you have cases that can provide additional scoring metrics beyond NRS (for pain) to further demonstrate improvements in ADL (activities of daily living) following TAME?
Response 2: Thank you for your advice. The following ADL scores are revised in the description: Quick-Dash Score of lateral epicondylitis and secondary stiff shoulder, PRWE score of TFCC injury, and FIHOA of finger OA, in [Page 4-8, Line 166-167, 205-207, 218-220, 279-280]. We apologize for not having the ADL scores for the cases of post-traumatic foot pain, patellar tendinopathy, and metatarsalgia.

Comments 3: The target vessels in TAME procedures are often very small, and the use of 4Fr guiding catheters, along with the complexity of the procedure, can sometimes result in vasospasm. Preventive strategies, such as using a 3Fr guiding catheter with a gentler approach and administering embolic material more proximally before reaching the target, have been suggested (Ref 3: J Vasc Interv Radiol, 2022. 33(12): p. 1468-1475.e8). Since you mention that approximately 70% of vasospasms occurred in the proximal part of the guiding catheter, it seems some cases could have been prevented by these techniques, potentially avoiding the need for NTG. This aspect should be discussed further in the manuscript.
Response 3: Thank you for the thorough and detailed review of the procedure method. In our cases, we use the 4Fr guiding catheter to engage the parent vessels. Then, we applied 1.7Fr. microcatheters for targeting vessel catheterization, and the vasospasms occurred in this step. Though the vasospasms were still noted on selective angiography caused by the 1.7Fr. microcatheter, A 3Fr guiding catheter may reduce the strength baseline stimulation in some cases and thus reduce the odds of vasospasm. Proximal microembolization was applied in some other cases in both of our instruments and provides good responses in most cases. However, we consider there are some remarkable benefits in selective microembolization such as evaluating the inflammatory neo vessels before and after the treatment, better reflexing the provoking pain during treatment, avoiding embolization toward the nontargeting vessels, and better dosage adjustment. We discuss these issues in our revised manuscripts on [Page 11, Paragraph 5.2, Line 352-361].

Comments 4: In the seventh case involving a stiff shoulder, did you perform any embolization beyond the right thoracoacromial artery? For conditions like frozen shoulder, embolization of six vessels around the shoulder is recommended (Ref 3: J Vasc Interv Radiol, 2022. 33(12): p. 1468-1475.e8). If the necessary vessels were not targeted, please provide additional details, as this might explain why TAME was ineffective in this case.
Response 4: Thank you, we appreciate your opinion. We do the embolization of thoracoacromial artery (TAA), circumflex scapular artery (CSA), suprascapular artery (SCA), posterior circumflex humeral artery (PCHA), anterior circumflex humeral artery (ACHA). The coracoid branch catheterization was not performed due to small orifice and branch, it might explain this case was ineffective. Furthermore, previous report (J Vasc Interv Radiol, 2021. 33(4): p. 489-496) showed still 16% of VAS pain >7at 6 months follow up in patients with secondary stiff shoulder (SSS) after embolization. The disease nature of SSS may be different to those of idiopathic chronic pains.

Reviewer 3 Report

Comments and Suggestions for Authors

1.      Hematology reports, Lipid profile and CRP value should be documented before and treatment.

2.      Additionally, blood urea, creatinine, sodium and potassium levels before and after treatment should be summarized in the form of a table.

3.      The main mechanisms of action behind the observed effects of NTG should be elaborately discussed.

4.      It is very essential to summarize all biochemical changes in the serum for possible adverse effects of NTG (Do not repeat physical parameters hypotension, dizziness ).

5.      Discussion is written very poor. Rewrite the discussion by inserting the cause of observed effects and discussion

6.      There is no new significant finding in this study. The current findings are already documented by previous investigators.

7.      I think that manuscript is written in hurry. There are many mistakes in the text like lack of space between two words, etc.

8.       The interaction of NTG with sodium chloride should be discussed I the infusion section.

9.      A brief but noticeable rise in blood pressure might result from a saline infusion. There are several potential causes for this, such as elevated nuclear factor kappa B synthesis and elevated angiotensin 1 receptor expression in renal tissues. NTG was diluted in saline before infusion. Therefore, authors should focus this point and discuss it in the discussion section.

Comments on the Quality of English Language

The place of comma, stop, and space between words should be considered. 

Author Response

Dear Reviewer,
Thank you very much for your insightful comments. I sincerely appreciate your feedback. Please see the attached PDF profile and our responses below:

Comments 1: Hematology reports, Lipid profile and CRP value should be documented before and treatment.
Response 1: Thak you for your precious advice. In clinical practice we don’t routinely acquire the lipid profile and CRP value for our TAME patients. We would consider acquiring and analysis these data in the following cases.

Comments 2: Additionally, blood urea, creatinine, sodium and potassium levels before and after treatment should be summarized in the form of a table.
Response 2: Thank you, we appreciate your advice. Some of the biochemistry indices reports are not routinely acquired. All patients were checked to have normal renal functions (eGFR > 90) for safety of transarterial angiography. [Page 2, Paragraph 2.1, Line77-78].

Comments 3: The main mechanisms of action behind the observed effects of NTG should be elaborately discussed.
Response 3: Thank you, we appreciate your opinion. The cause of the observed effects is due to the mechanism of NTG acting on smooth muscle, as revised and shown in [Page 10, Paragraph 5.1, Line 330-333].

Comments 4: It is very essential to summarize all biochemical changes in the serum for possible adverse effects of NTG (Do not repeat physical parameters hypotension, dizziness).
Response 4: Thak you for point this out. The biochemical mechanism of NTG acting on smooth muscle was revised and shown in [Page 10, Paragraph 5.1, Line 330-333].

Comments 5: Discussion is written very poor. Rewrite the discussion by inserting the cause of observed effects and discussion
Response 5: Thak you for your precious advice. We have made a discussion about proximal catheter stenosis and alternative nontargeting or simplified microembolization in [Page 11, Paragraph 5.2, Line 352-361].

Comments 6: There is no new significant finding in this study. The current findings are already documented by previous investigators.
Response 6: Thak you for your precious advice. Based on our literature review, there are reports about the vasodilating effect of NTG in hand angiography, as shown in (Ref 11: J Vasc Interv Radiol, 2003. 14(6): p. 749-754). However, TAME is a relatively new procedure in chronic pain treatment and application of NTG infusion in TAME is not formally reported.

Comments 7: I think that manuscript is written in hurry. There are many mistakes in the text like lack of space between two words, etc.
Response 7: Thak you for your recommendation. We have repaired the typographical error in [Line 51 and 52: "as" in figure 1 and 2], and an extra space in [Line 93].

Comments 8: The interaction of NTG with sodium chloride should be discussed I the infusion section.
Response 8: Thank you for pointing this out. The NTG dosage we used in this report is rather low (50-100mcg), as shown in [Page 4, Paragraph 2.10, Line 149-150]. Also, the interaction is not observed in previous reports in (Ref 16: Eur Radiol 2012. 22(10): p.2193-2200).

Comments 9: A brief but noticeable rise in blood pressure might result from a saline infusion. There are several potential causes for this, such as elevated nuclear factor kappa B synthesis and elevated angiotensin 1 receptor expression in renal tissues. NTG was diluted in saline before infusion. Therefore, authors should focus this point and discuss it in the discussion section.
Response 9: We appreciate your opinion. The angiography is performed with heparinized saline and iodine contrast agent. The volume of saline in NTG, less than 2ml, as shown in [Page 4, Paragraph 2.10, Line 149-150], is far less than the saline used in TAME and other angiography cases. Thus, the blood pressure change due to saline infusion in the NTG administration would be masked by the normal TAME and angiography procedure.
